# Skewed X-inactivation is associated with retinal dystrophy in female carriers of *RPGR* mutations

Muhammad Usman[1], Christoph Jüschke[1], Fei Song[1], Dennis Kastrati[1], Marta Owczarek-Lipska[1,2], Jannis Eilers[1], Laurenz Pauleikhoff[3], Clemens Lange[3,4], John Neidhardt[1,5]

**Progressive degeneration of rod and cone photoreceptors frequently is caused by mutations in the X-chromosomal gene Retinitis Pigmentosa GTPase Regulator (*RPGR*). Males hemizygous for a *RPGR* mutation often are affected by Retinitis Pigmentosa (RP), whereas female mutation carriers only occasionally present with severe RP phenotypes. The underlying pathomechanism leading to RP in female carriers is not well understood. Here, we analyzed a three-generation family in which two of three female carriers of a nonsense *RPGR* mutation presented with RP. Among two cell lines derived from the same female family members, differences were detected in *RPGR* transcript expression, in localization of RPGR along cilia, as well as in primary cilium length. Significantly, these differences correlated with alterations in X-chromosomal inactivation patterns found in the patient-derived cell lines from females. In summary, our data suggest that skewed X-chromosomal inactivation is an important factor that determines the disease manifestation of RP among female carriers of pathogenic sequence alterations in the *RPGR* gene.**

## Introduction

Rod and cone photoreceptors are essential for vision and phototransduction. Retinitis pigmentosa (RP, OMIM #268000), also known as Retinopathia Pigmenti or rod-cone dystrophy, is an inherited condition caused by progressive degeneration predominantly of rod photoreceptors. Symptoms of RP include night blindness and peripheral vision loss which eventually may lead to complete blindness over several decades of disease progression. The disease is estimated to affect 1 in 4,000 people globally (Hartong et al, 2006). RP may either be non-syndromic (affecting only photoreceptors) or syndromic (diagnosis such as Bardet-Biedl syndrome, Usher syndrome) (Katsanis, 2004; Keats & Savas, 2004). To date, over 70 genes have been mapped or identified to be associated with non-syndromic RP (RetNet™, 2022). The majority of RP cases are affected by autosomal dominant, autosomal recessive, or X-linked forms of the disease. Indeed, X-linked non-syndromic RP (XLRP) is among the most severe forms of RP and accounts for ~15% of the disease burden (Daiger, 2004; Daiger et al, 2007).

*RPGR* [OMIM312610] is a gene located on the short arm of the X-chromosome (Xp21.1). Mutations in the *RPGR* gene cause up to 80% of XLRP cases (Roepman et al, 2000; Vervoort et al, 2000; Pelletier et al, 2007; Tee et al, 2016; Stone et al, 2017). The RPGR protein predominantly localizes to the connecting cilium of the rod and cone photoreceptors and along the axoneme of primary cilia (Hong et al, 2003; Da Costa et al, 2015). Primary cilia are antenna-like protrusions on the surface of almost all types of cells and serve as signaling hubs for intra- and extracellular signaling (Pazour & Witman, 2003). The connecting cilium of photoreceptors resembles primary cilia and may have developed specialized functions in photoreceptor-dependent protein sorting and trafficking (Hong et al, 2003; Da Costa et al, 2015). RPGR ciliary protein localization seems to be crucial for its function. Mislocalization of RPGR was associated with disease manifestations (Da Costa et al, 2015).

XLRP is caused by a mutation in the *RPGR* gene and affects males in particular. Nevertheless, females can also develop the disease phenotype displaying a broad phenotypic spectrum from mildly to severely affected (Jin et al, 2007). The molecular mechanism underlying the phenotypic variability in the female carriers of XLRP is not well understood; it has been associated with X-chromosome inactivation (XCI) (Banin et al, 2007; Fahim & Daiger, 2016). Notably, XCI is a cellular mechanism in females that balances the expression of X-linked genes by randomly inactivating one of the two X-chromosomes at early embryonic stages (Lyon, 2002). Later on, all the descendant cells will have the same inactivated parental X-chromosome (van den Berg et al, 2009). Skewing of XCI ratios, clearly diverging from a 50:50 ratio, has been associated with disease manifestation in female carriers of e.g., Duchenne muscular dystrophy, hemophilia, and XLRP (Pegoraro et al, 1994; Di Michele et al, 2014; Fahim & Daiger, 2016).

[1]Human Genetics, Medical Faculty, School of Medicine and Health Sciences, Carl von Ossietzky Universität Oldenburg, Oldenburg, Germany   [2]Junior Research Group, Genetics of Childhood Brain Malformations, School of Medicine and Health Sciences, University of Oldenburg, Oldenburg, Germany   [3]Eye Center, Medical Center – University of Freiburg, Faculty of Medicine, University of Freiburg, Freiburg im Breisgau, Germany   [4]Ophtha-Lab, Department of Ophthalmology at St. Franziskus Hospital, Muenster, Germany   [5]Research Center Neurosensory Science, Carl von Ossietzky University Oldenburg, Oldenburg, Germany

Correspondence: john.neidhardt@uni-oldenburg.de

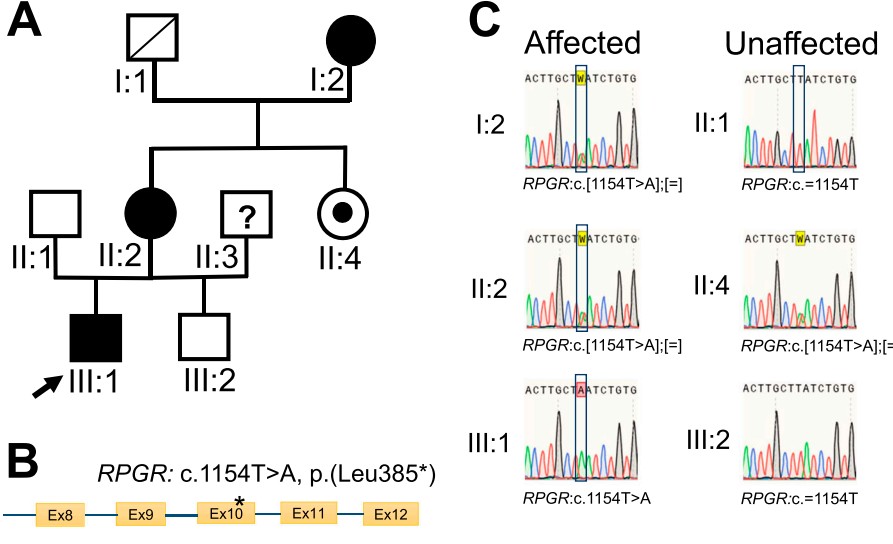

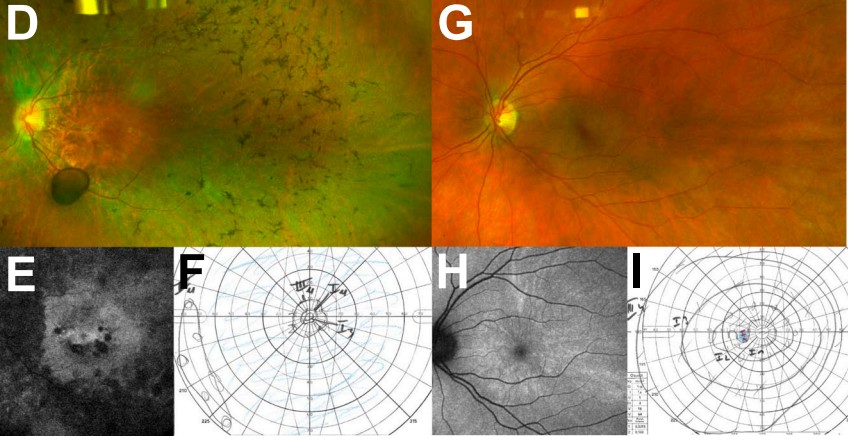

**Figure 1. Genotype and phenotype analyses.**
**(A)** A three-generation pedigree segregating RP. Males are shown by square symbols, females are shown by circles. The filled symbols denote individuals affected by RP. The index patient is marked by an arrow. The dot in the middle of the circle indicates an asymptomatic carrier in the family. The deceased individual is denoted by an angled line within the square. The question mark indicates that no further information from this individual is available. **(B)** Schematic diagram of the *RPGR* gene showing the position of the pathogenic variant in exon 10. **(C)** Sanger sequencing was applied to analyze the pathogenic variant *RPGR*: c.1154T>A from genomic DNA of blood samples from all participants of the study. Sequence electropherograms of affected individuals (I:2, II:2, and III:1) and unaffected individuals (II:1, II:4, and III:2) of the family are shown. **(D, E, F)** Fundoscopy shows bone spicules as well as constricted retinal vessels in patient II.2. FAF imaging revealed large areas of retinal pigment epithelial (RPE) atrophy. Visual field testing showed a constriction of isopter III.4 to about 10°. **(G, H, I)** The unaffected carrier II:4 (sister of II:2) showed an unremarkable funduscopic exam and visual field. Her FAF image showed no RPE atrophy but a pattern of radially extending hyperautofluorescence that has previously been described in female carriers of *RPGR* mutations.

During development, random inactivation of one X-chromosome in an early embryonic stage may lead to XCI patterns that differ among tissues or even within the same tissue. Thus, selecting a suitable tissue to determine XCI patterns in female carriers is challenging (Cotton et al, 2011). As retinal tissues are not available to determine retinal XCI patterns in female carriers of an *RPGR* pathogenic sequence alteration, blood and/or saliva tissues are often used to determine XCI patterns. However, blood and saliva share ~70% of white blood cells as a common source of DNA and may not provide complementary information. Previously, buccal mucosa and skin tissues have been associated with more reliable XCI pattern information in female carriers (de Hoon et al, 2015; Fahim et al, 2020). The androgen receptor gene (*AR*) is widely used to determine the XCI status within a patient sample (Allen et al, 1992). In addition, retinitis pigmentosa 2 (*RP2*), SLIT and NTRK-like family member 4 (*SLITRK4*), and proprotein convertase subtilisin/kexin type 1 inhibitor (*PCSK1N*) genes were identified as XCI markers (Bertelsen et al, 2011; Machado et al, 2014).

Clinical trials of gene therapies to treat *RPGR* pathogenic variants are currently underway but mostly include male patients (Cehajic-Kapetanovic et al, 2020). Consequently, it is essential to understand the molecular pathomechanisms underlying XLRP in female carriers to be able to optimize gene treatments.

In this family-based study, we investigated female carriers with *RPGR*-associated XLRP who shared a pathogenic variant and a similar genetic background. We found that skewed XCI is associated with ciliary mislocalization of the RPGR protein, with shortening of primary cilia, as well as with the clinical diagnosis of RP.

## Results

### A pathogenic nonsense variant in exon 10 of the *RPGR* gene is associated with RP in females

The three-generation family (Fig 1A) included an index male patient (III:1), two affected female carriers (I:2 & II:2), one asymptomatic carrier female (II:4), and two unaffected male relatives (II:1 and III:2). The index patient is severely affected by RP. Previously, we showed that the male index patient carries the nonsense pathogenic sequence variant *RPGR*: c.1154T>A in exon 10 (NM_000328.3) (Fig 1B)

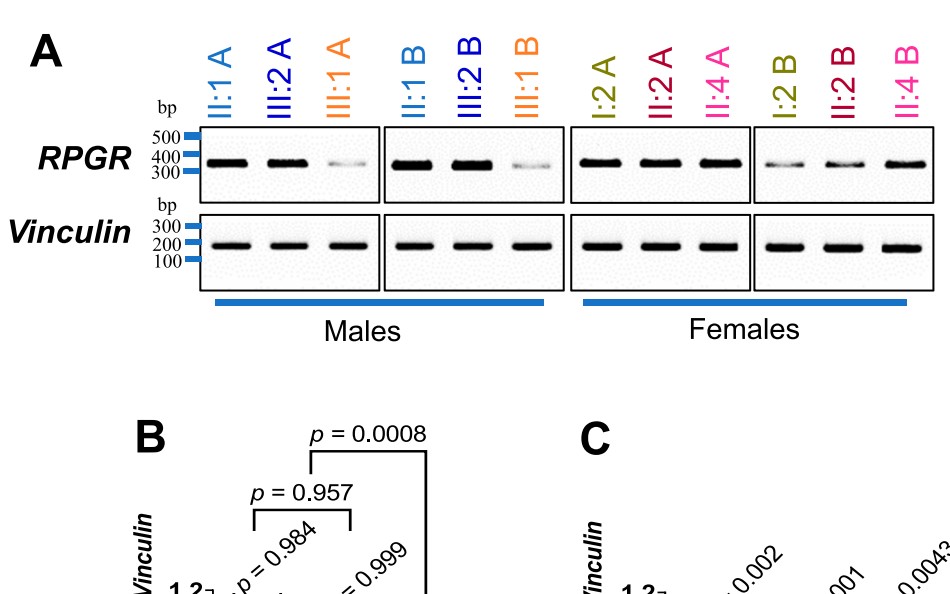

**Figure 2.   *RPGR* transcript analyses in two independent cell lines.**
**(A, B)** *RPGR* transcripts between exon 2 through exon 5 were amplified by RT–PCR in two independently generated cell lines (A, B) from each healthy relative (II:1, III:2, II:4) and each patient (III:1, I:2, II:2). DNA marker sizes are given as base pairs (bp). *Vinculin* was used as a loading control. **(A, B)** Semi-quantitative assessment of *RPGR* transcripts relative to *vinculin* in two fibroblast cell lines (A, B) derived from the index patient (III:1) as well as from healthy male relatives (II:1, III:2). *P*-values were calculated by one-way ANOVA. **(A, B, C)** Semi-quantitative assessment of *RPGR* transcripts relative to the loading control *vinculin* in two cell lines (A, B) derived from each female relative (I:2, II:2, II:4). The affected females (I:2, II:2) and the asymptomatic female (II:4) are heterozygous carriers for the pathogenic sequence alteration *RPGR*: c.1154T>A. *P*-values were calculated by one-way ANOVA. Experiments were independently repeated three times; error bars indicate SD; non-significant: *P* ≥ 0.05; significant: *P* < 0.05.

(Vossing et al, 2020). Herein, we analyzed additional family members and found that all three females in the pedigree are carriers of the *RPGR* pathogenic alteration (*RPGR*: c.[1154T>A];[=]) (Fig 1C). Consequently, the nonsense pathogenic variant in exon 10 of the *RPGR* gene was associated with RP also in females.

The affected female II:2 showed a reduced best corrected visual acuity (BCVA) of 20/50 on her right and 20/100 in her left eye. Her visual field was constricted to about 10° (isopter III.4) on both eyes, and fundus autofluorescence (FAF) imaging revealed a widespread area of retinal pigment epithelial atrophy (Fig 1D–F). Full-field electroretinography (ffERG) and multifocal electroretinogram (mfERG) both showed almost extinguished responses. The unaffected female II:4, by contrast, showed a BCVA of 20/20 in her right and 20/15 in her left eye. Her visual fields and mfERG were unremarkable. Skotopic ffERG responses tended towards lower responses, while photopic responses were within normal limits. Her FAF images did not reveal retinal pigment epithelial atrophy but were consistent with a radial pattern of asymptomatic carriers that have previously been described (Fig 1G–I) (Nanda et al, 2018).

Skewed XCI might be the reason for the phenotypic variability in female carriers of the pathogenic variant *RPGR*: c.1154T>A. To understand the contribution of XCI in the phenotype variability in female carriers, we generated two independent fibroblast cell lines (A and B) from two different skin biopsies of each of the six family members that participated in the study (I:2, II:1, II:2, II:4, III:1, and III:2). Importantly, I:2 and II:2 are affected female carriers, whereas I:4 is

an unaffected carrier of the pathogenic variant. In addition, two control cell lines from unrelated healthy female donors were included in the study (Figs S1–S3). These healthy individuals did not report signs of visual impairment and did not show sequence alterations in genes associated with retinal dystrophy. Sanger sequencing confirmed the genotypes of control and patient-derived fibroblast cell lines (both A and B) including affected and unaffected participants (Figs 1C and S1C).

### Transcript analyses from female carriers of an *RPGR* pathogenic variant

We previously found that the nonsense-mediated mRNA decay (NMD) pathway degrades the mutated *RPGR* transcript in a patient-derived fibroblast cell line of the index patient (III:1) (Vossing et al, 2020). Herein, we confirmed this observation in a second, independently generated cell line from the index patient (Fig 2A and B). Furthermore, two independent cell lines of each female carrier (heterozygous for the *RPGR* pathogenic variant c.1154T>A) were analyzed. We found that both cell lines (denoted A and B) of affected female carriers (I:2 and II:2) showed differences in *RPGR* transcript expression, in which A exhibited the reference levels of *RPGR* transcripts and B was significantly lower (Fig 2A and C). This supported that degradation by NMD (Fig 2A and C) occurred to a different extent among the two independently generated cell lines derived from the same individual. Moreover, both cell lines A and B

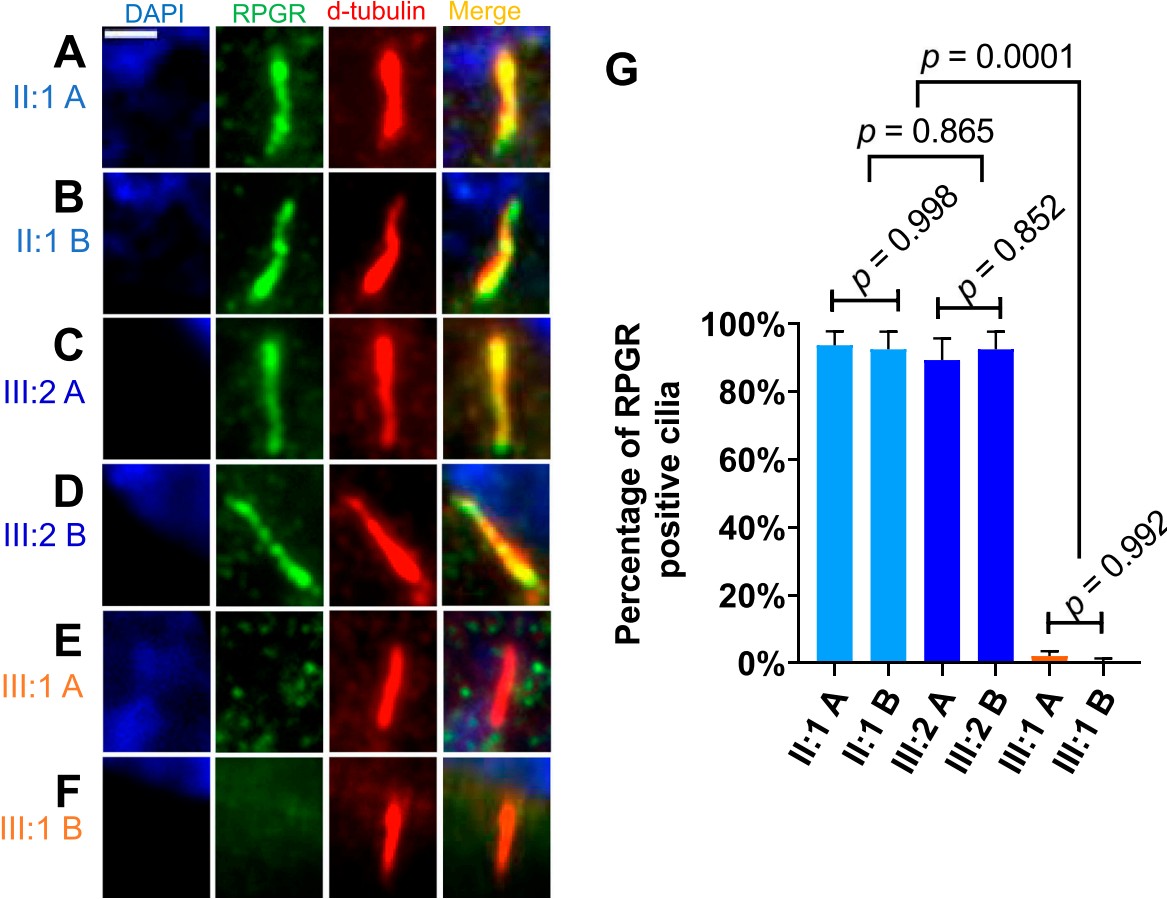

**Figure 3. Localization of RPGR proteins along primary cilia in cell lines derived from two healthy male family members (II:1 and III:2) as well as from the male index patient (III:1).**
**(A, B)** Two independently generated fibroblast cell lines (A or B) from each individual were analyzed. **(A, B, C, D, E, F)** Immunofluorescence detection of RPGR ciliary localization. The cell nucleus was stained with DAPI (blue), RPGR was detected with an anti-RPGR antibody (green), and the ciliary axoneme was identified with an anti-d-tubulin antibody (red). **(G)** Quantitative assessment of RPGR protein localization along primary cilia. *P*-values were calculated by one-way ANOVA. Experiments were independently repeated five times. 50 cilia from each independent experiment were quantified. Error bars indicate SD; non-significant: $P \geq 0.05$; significant: $P < 0.05$. Scale bar: 2 $\mu$m.

of the asymptomatic carrier female also showed significant differences in *RPGR* transcript levels, although the differences were less clear than in the affected carriers (Fig 2A and C). Fibroblast cell lines from two unrelated female controls were found to express *RPGR* transcripts at the reference level (Fig S1A and B). Moreover, Sanger sequencing of cDNAs generated from *RPGR* transcripts (extracted from fibroblasts) revealed preferences to express a single allele. Only the reference sequence was detectable in both A cell lines of affected females (I.2 A and II:2 A), resembling the sequencing result of unaffected males (Fig S1C). In contrast, the B cell lines of the two affected female carriers predominantly expressed mutated *RPGR* transcripts, thus resembling the index patient (Fig S1C). In summary, our findings indicate that female carriers of *RPGR* pathogenic variants may exhibit detectable differences between two independently generated cell lines from the same individual.

### Ciliary localization of RPGR is disturbed in the index patient

RPGR protein localization along primary cilia is considered essential for its function (Vossing et al, 2020). We detected the RPGR

proteins along primary cilia in fibroblasts by immunocytochemistry using an antibody directed against RPGR.

In males, the healthy family members (II:1 and III:2) showed RPGR along the axoneme in the majority of fibroblast-derived cilia (Fig 3A–D). In contrast, we did not detect RPGR in cilia from the index patient (III:1) (Fig 3E and F). We quantified the RPGR localization along primary cilia and confirmed these observations. Significant differences between patient and control cell lines were detected (Fig 3G). This dataset supports that the *RPGR* nonsense pathogenic variant c.1154T>A prevented ciliary localization of RPGR proteins in patient-derived fibroblast cell lines (both A and B) of the index male patient (III:1).

### Female carriers of pathogenic *RPGR* sequence alterations may show differences in the localization of ciliary RPGR protein

In female carriers of pathogenic sequence alterations, we asked whether RPGR shows differences in primary cilia localization. We analyzed two independently generated fibroblast cell lines (A and B) from each female carrier of the pathogenic *RPGR* variant (I:2, II:2,

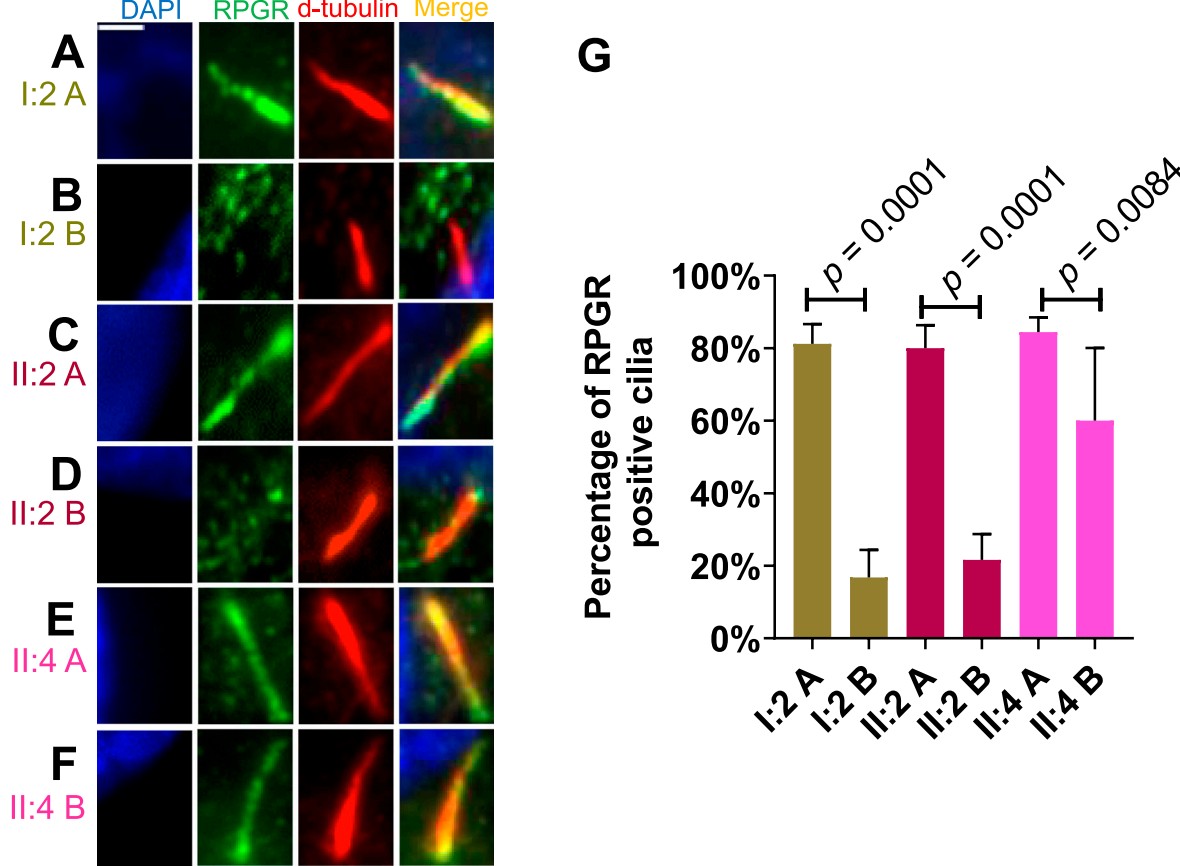

**Figure 4. Localization of RPGR protein along primary cilia in female carriers of a pathogenic *RPGR* sequence variant.**
Ciliary RPGR localization was evaluated in female carriers (I:2, II:2, and II:4) of the *RPGR* pathogenic variant *RPGR*: c.1154T>A. **(A, B)** We analyzed two independently generated cell lines (denoted A, B) of the same individual. **(A, B, C, D, E, F)** Immunofluorescence detection of RPGR along the axoneme of primary cilia. The cell nucleus was stained with DAPI (blue), RPGR was detected with an anti-RPGR antibody (green), and the ciliary axoneme was marked by an anti-d-tubulin antibody (red). **(G)** Quantitative assessment of RPGR localization along the primary cilium in female carriers of the pathogenic *RPGR* sequence alteration *RPGR*: c.1154T>A (I:2, II:2, and II:4). *P*-values were calculated by one-way ANOVA. Experiments were independently repeated five times. 50 cilia from each independent experiment were quantified. Error bars indicate SD; non-significant: $P \geq 0.05$; significant: $P < 0.05$. Scale bar: 2 $\mu$m.

and II:4) (Fig 4) as well as from two unrelated healthy female controls (Fig S2A and B). We found that the A cell lines of the two affected female carriers (I:2 A and II:2 A) showed normal RPGR localization along the primary cilia (Figs 4A and C and S2A and B). Significantly, RPGR along primary cilia frequently was not detectable in cell line B derived from the same affected female carriers (I:2 and II:2) (Fig 4B and D). In addition, the comparison of the two cell lines A and B derived from the asymptomatic carrier female (II:4) showed differences in the localization of RPGR along the axoneme, although these differences were less clear than in the symptomatic female family members (I:2 and II:2) (Fig 4E and F). In two unrelated female controls, we observed normal RPGR ciliary localization (Fig S2A and B). We quantified these findings and detected that cell line A of the two affected female carriers (I:2 A and II:2 A) showed RPGR along the primary cilia in ~80% of the cells (Fig 4G). In contrast, the second patient-derived fibroblast cell lines (B) of I:2 and II:2 exhibited only about 20% cilia with RPGR (Fig 4G). The two cell lines A and B of the asymptomatic female carrier (II:4 A and II:4 B) showed smaller but detectable variances of ~20% in RPGR-positive cilia (Fig 4G). Of note, II:2 B was significantly different from II:4 A. In summary,

our data demonstrate that two fibroblast cell lines derived from the same female carrier of an *RPGR* pathogenic variant can exhibit differences in RPGR ciliary localization.

## Primary cilia length was associated with transcript levels and ciliary localization of RPGR

We compared cell lines derived from healthy and affected relatives to understand whether the loss or reduction in RPGR localization along cilia influences primary cilia length. The index patient and his healthy male relatives as well as female carriers of the *RPGR* pathogenic variant were included in this comparison. We found that all cell lines can generate primary cilia (Fig 5A–L). We measured the length of the axoneme and compared all cell lines derived from the family. Two unrelated healthy female controls were also included (Fig S3). In the male index patient III:1, we found that the primary cilium length was significantly reduced in both fibroblast cell lines (Fig 5M). We further detected that the two cell lines of each affected female carrier (I:2 and II:2) exhibit different axoneme lengths (Fig 5N) (Fig S3B), a finding that closely correlated with the loss/

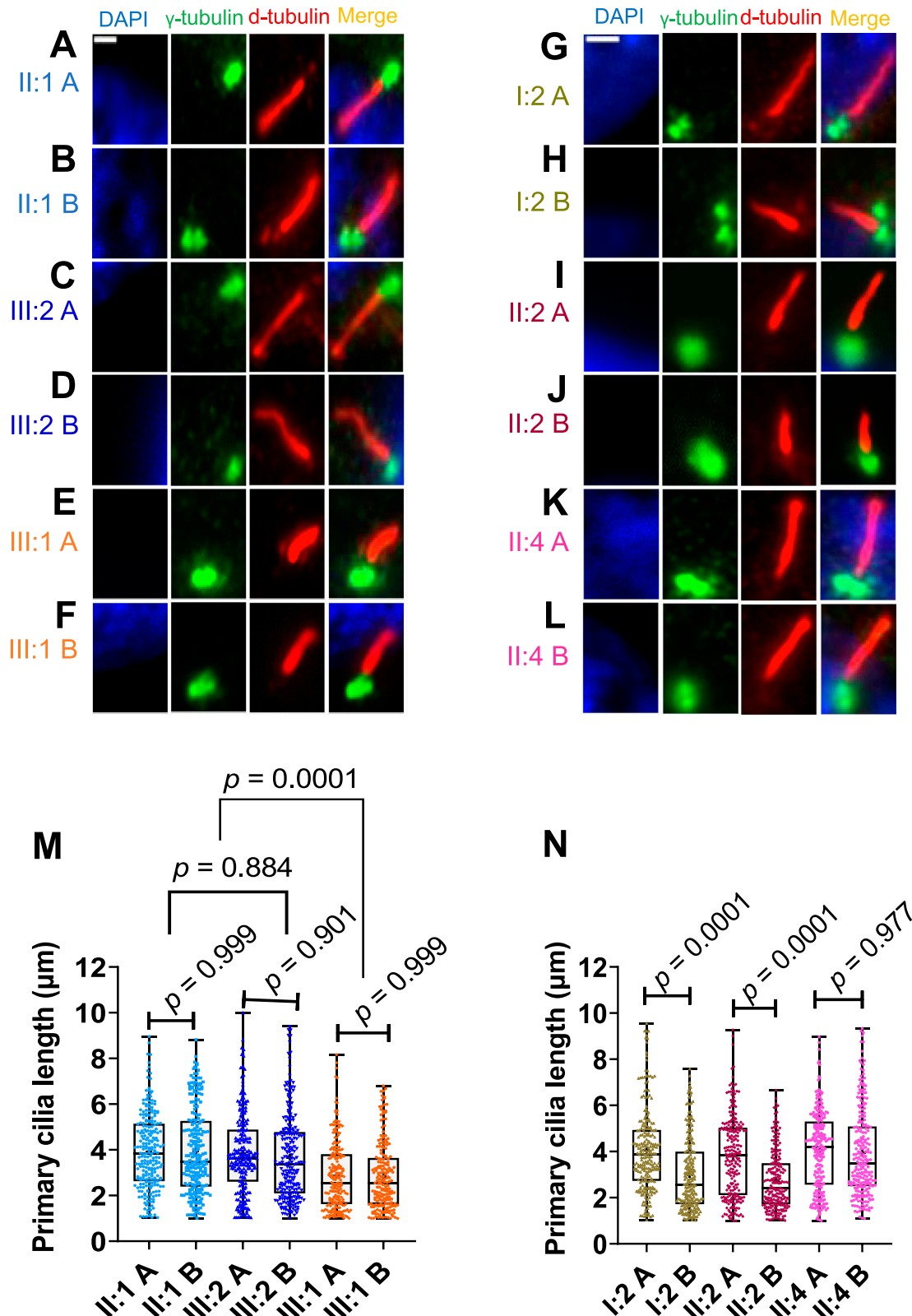

**Figure 5. Length measurements of primary cilia axonemes in control and patient-derived cell lines.**
We analyzed healthy male relatives (II:1 and III:2), the index patient (III:1), and the female carriers (I:2, II:2, and II:4) of the *RPGR* pathogenic variant *RPGR*: c.1154T>A.
**(A, B, C, D, E, F, G, H, I, J, K, L)** Immunofluorescence detection of primary cilia. The cell nucleus was stained with DAPI (blue). The basal body was detected with an anti-γ-tubulin antibody (green). The ciliary axoneme was marked with an anti-d-tubulin antibody (red). **(A, B, M, N)** Quantitative assessment of primary cilia length in two

reduction of *RPGR* transcripts and with the mislocalization of RPGR along primary cilia (Figs 3G and 4G) (Figs S1A and B and S2B). Although we did not detect significant differences between cell lines A and B of the asymptomatic carrier female II:4, cell line B displayed a tendency towards shorter cilia (Fig 5N). In summary, our data strongly suggest that RPGR is crucial for the maintenance and/or stability of primary cilia and that its ciliary localization closely correlates with ciliary length alterations.

### XCI is a modulator of RP in females carrying a pathogenic *RPGR* variant

To verify whether a non-random XCI may explain the differences in RPGR protein localization along primary cilia, we analyzed the methylation status between the two X-chromosomal alleles in female family members. We compared cell lines A and B of female carriers of the *RPGR* pathogenic variant and analyzed four loci on the X-chromosome that have previously been described as suitable markers of altered XCI.

We found that cell line A of the affected carrier female I:2 showed skewed methylation of X-chromosomal alleles, which is causing the predominant expression of the reference-*RPGR* allele (Table 1). This was in contrast with her cell line B which showed a skewed expression towards the mutated *RPGR* allele (Table 1). In the second affected carrier female II:2, cell line A showed random XCI, whereas cell line B was skewed towards mutated-*RPGR* expression (Table 1). The asymptomatic carrier of the pathogenic *RPGR* variant (II:4) presented with skewed expression towards the reference-*RPGR* allele in cell line A and a preferred activity of the reference-*RPGR* allele in cell line B. XCI ratios of fibroblast cell lines A and B, derived from carriers of the pathogenic *RPGR* variant, remained consistent throughout multiple passages in the cell culture (Table S1). Significantly, the XCI data from the individual fibroblast cell lines are in accordance with the results described above, including the *RPGR* transcript analyses, the cDNA sequencing, the ciliary localization of RPGR proteins, and the ciliary length measurements (also compare to Table S2). The XCI status of at least one of the individual cell lines correlated with the clinical phenotype of the female carriers.

It is unclear why two out of eight values from individual II:2 (*AR* and *PCSK1N*) suggested XCI ratios that differed from the other loci (Table 1). In addition, the XCI status in blood tissue-derived genomic DNA was a discrepancy with the results from fibroblast cell lines. Indeed, we did not find a correlation between blood-XCI ratios and other data presented herein, a finding that also included the clinical phenotype of females carrying the *RPGR* pathogenic sequence alteration (Table 1 and Fig 1D–F).

In conclusion, our data showed that the XCI ratios detected in fibroblasts closely correlate with functional properties of RPGR, strongly suggesting the relevance of XCI for the clinical variability in RP phenotypes in females that carry a pathogenic *RPGR* alteration. Genomic DNA from peripheral blood seems less well suited to correlate individual XCI statuses with the clinical phenotype.

## Discussion

This study describes a family in which a pathogenic *RPGR* variant causes severe RP in male and female members. We demonstrate that the mutation in *RPGR* leads to severe primary cilia defects resulting in the shortening of the axoneme length in patient-derived fibroblast cell lines. To the best of our knowledge, we showed for the first time that skewed X-chromosomal inactivation and RPGR-related ciliary defects are closely associated. Thus, our data support that the disease manifestations in female carriers of pathogenic *RPGR* variants are modulated by misbalanced XCI. We further reported that analyzing a single patient-derived tissue sample, e.g., a blood sample, as well as analyzing a single XCI marker conveys a high risk for false positive or false negative results.

The pathogenic variant *RPGR*: c.1154T>A is predicted to affect essential RPGR protein domains, i.e., the RCC1-like domain, and to cause a truncation of the protein including the isoprenylation motif (CAAX) at the C-terminus. Mutations in the RCC1-like domain region of the RPGR tend to cause more severe RP phenotypes (Meindl et al, 1996; Wright et al, 2011). In addition, the nonsense pathogenic variant reduced the *RPGR* transcript levels significantly, an observation that likely is because of NMD. Although the residual transcript levels affected by the mutation might still translate into a truncated protein, it seems unlikely that this would result in functional RPGR proteins. It is unclear whether NMD or the stop in protein translation is the main pathomechanisms underlying the *RPGR* pathogenic variant. The absence of detectable RPGR proteins in the primary cilium indicates that the mutated protein either is degraded or cannot enter the primary cilium. Notably, the CAAX motif at the C-terminus of the RPGR protein is known to be relevant for ciliary localization (Zhang et al, 2019). In conclusion, our data support that the *RPGR* pathogenic variant likely is a functionally null mutation that causes defects in ciliary properties associated with retinal diseases as observed in the family described herein.

XLRP particularly affects males. Occasionally, females can present with an XLRP phenotype, an observation for which the reasons are not yet fully understood. Female carriers of *RPGR* pathogenic variants present a broad clinical spectrum that may vary from unaffected to severely affected. In a retrospective study of 23 female *RPGR* carriers, four different classifications based on AF images were recently described to distinguish carriers with pathogenic *RPGR* variants. In addition to a normal appearance (4% of cases), a radial reflex pattern without pigmentary retinopathy (61%), focal pigmentary retinopathy (22%), and a severe RP phenotype comparable to that presented by males (13%) were described (Nanda et al, 2018). The clinical presentations of the two affected female patients I:2 and II:2 described in this study are best described by the category of severe RP, whereas the unaffected female II:4 shows only radial reflex pattern without additional signs of retinal or visual alterations (for representative examples of retinae from unaffected individuals, compare to Zhang et al [2021]). Why female carriers of pathogenic *RPGR* variants exhibit XLRP with

independent cell lines (denoted A, B) from each individual. We analyzed healthy individuals (II:1, III:2), the affected index patient (III:1), and female carriers of the *RPGR* pathogenic variant (I:2, II:2, and II:4). *P*-values were calculated by one-way ANOVA. In total, the axonemes of 210–250 cilia from five independent experiments (40–50 cilia each) were measured using Fiji-ImageJ software. Non-significant: $P \geq 0.05$; significant: $P < 0.05$; scale bar: 2 $\mu$m.

**Table 1. X-chromosomal inactivation in fibroblast- and blood-derived genomic DNA.**

| Loci | DNA resource: Fibroblast cell lines | | | | | | DNA resource: Blood | | |
|---|---|---|---|---|---|---|---|---|---|
| | I:2 A | I:2 B | II:2 A | II:2 B | II:4 A | II:4 B | I:2 | II:2 | II:4 |
| | Methylation status, alleles mut-*RPGR*: Ref-*RPGR* | Methylation status, alleles mut-*RPGR*: Ref-*RPGR* | Methylation status, alleles mut-*RPGR*: Ref-*RPGR* | Methylation status, alleles mut-*RPGR*: Ref-*RPGR* | Methylation status, alleles mut-*RPGR*: Ref-*RPGR* | Methylation status, alleles mut-*RPGR*: Ref-*RPGR* | Methylation status, alleles mut-*RPGR*: Ref-*RPGR* | Methylation status, alleles mut-*RPGR*: Ref-*RPGR* | Methylation status, alleles mut-*RPGR*: Ref-*RPGR* |
| *AR* | 96:04 (±2) | 11:89 (±5) | 48:52 (±7) | 49:51 (±4)[a] | 93:07 (±4) | 83:17 (±5) | 16:84 | 48:52 | 71:29 |
| *RP2* | 95:05 (±2) | 16:84 (±2) | 59:41 (±4) | 18:82 (±1) | 77:23 (±6) | 60:40 (±9) | 20:80 | 44:56 | 68:32 |
| *SLITRK4* | 82:18 (±4) | 20:80 (±2) | 45:55 (±5) | 19:81 (±3) | 91:09 (±9) | 73:27 (±8) | 14:86 | 40:60 | 70:30 |
| *PCSK1N* | 90:10 (±2) | 15:85 (±6) | 70:30 (±3)[a] | 16:84 (±2) | 89:11 (±2) | 86:14 (±15) | 17:83 | 35:65 | 58:42 |

The table provides the ratios of methylation between the two X-chromosomal alleles of female carriers of the pathogenic sequence alteration c.1154T>A. The methylation of X-chromosomal alleles (encoding *RPGR*) was considered skewed towards either allele if ratios of XCI were ≥80:20. It was considered moderately skewed if the ratios were between 80:20 and 70:30. Random XCI was detected with ratios between 50:50 and 70:30. Values are given as mean (±SD) from at least three independent experiments. We included a healthy (II:4) and two RP-affected individuals (I:2 and II:2). Genomic DNA from either peripheral blood or two fibroblast cell lines (A and B) of each female carrier was analyzed. One allele carries the mutated *RPGR* (mut-*RPGR*), and the other allele carries the reference *RPGR* (ref-*RPGR*). Polymorphic regions from androgen receptor (*AR*), retinitis pigmentosa 2 (*RP2*), SLIT and NTRK-like family member 4 (*SLITRK4*), and proprotein convertase subtilisin/kexin type 1 inhibitor (*PCSK1N*) were used as markers for XCI.
[a]The values do not seem to correlate with the values of other XCI markers from the same fibroblast cell line.

varying clinical severity is currently the subject of debate and may be related to either additional genetic modifiers or skewed XCI (Fahim & Daiger, 2016; Fahim et al, 2020; Salvetti et al, 2021; Bhat et al, 2022).

Our cellular models supported a correlation between XCI defects and disease expression in female carriers of a pathogenic *RPGR* variant. The molecular and ciliary findings suggested that a random distribution of XCI protects against cellular phenotypes in carriers of a pathogenic *RPGR* variant (also compare to Table S2). In contrast, a skewed XCI towards the mutated allele causes RPGR defects and ciliary phenotypes also in females (Table S2). Notably, our Sanger sequencing analyses of transcripts expressed in female patient- and control-derived fibroblasts may suggest that misbalanced XCI leads to the expression of either only reference or only mutated transcripts. This observation is likely biased by the detection limits of the Sanger sequencing technology and the fact that PCR-based methods often prefer the more abundant templates; thus, we speculate that the other allele is not completely silenced.

Even though *AR* has been used as a standard XCI marker for several studies, including retinal dystrophies (Allen et al, 1992; Fahim et al, 2020; Wang et al, 2021), we herein suggest that several XCI markers should be compared with minimize the risk of false positive or false negative results. In line with this, we observed that the *AR* marker in cell line B of the affected carrier female II:2 did not follow the other three XCI markers. The reason for this is unknown and requires additional analyses. Furthermore, genomic DNA from peripheral blood showed little correlation with the phenotype of female carriers of the pathogenic *RPGR* variant. This is in agreement with studies on Rett syndrome suggesting that blood-derived genomic DNA is not a reliable resource to determine the XCI status of female carriers (Xiol et al, 2019). Moreover, previous studies have demonstrated an age-related increase in XCI skewing in blood samples (Busque et al, 2009), while no significant association between XCI variations and age has been observed in the skin (Zito et al, 2019). In our study, the female participants had the following ages: I:2 (60 yr), II:2 (42 yr), and II:4 (40 yr). Our investigations of skin-derived cell lines from

these carrier females showed consistent XCI ratios over several passages in culture (Table S1). However, our study did not longitudinally track the female carriers over time; hence, we cannot exclude that XCI ratios in the skin may vary with the age of the donor.

A study conducted by Vianna et al demonstrated that single nucleotide variants in four X-linked genes (*NLGN4X*, *USP9X*, *HDAC8*, and *TAF1*) have the potential to influence XCI skewing (Vianna et al, 2020). Although we did not detect any of these XCI-associated single nucleotide variants, it cannot be fully excluded that additional sequence variants influenced the misbalanced XCI ratios detected in our study. Further studies are required to evaluate whether genetic predispositions may influence XCI ratios in female patients with RP.

There is an ongoing debate about whether skewed XCI should be defined as a ratio of ≥80:20 or ≥90:10 (Fahim et al, 2020; González-Ramos et al, 2020). In our analysis, we applied the ≥80:20 ratio to define skewed XCI. This decision was justified based on our dataset and included analyses on transcript, protein, and cellular levels. Nonetheless, it remains an intriguing prospect for future studies to investigate and determine which threshold is more representative and clinically relevant in assessing XCI skewing.

Together, our data suggested that the loss of RPGR function impacts ciliary properties, which, in turn, correlated with XLRP in female carriers of pathogenic *RPGR* variants. The loss of RPGR function likely is modulated by the XCI ratios of female carriers. Our study expands the understanding of the pathomechanisms underlying pathogenic *RPGR* variants and will help optimize gene therapeutic applications toward treating *RPGR* defects in affected female carriers.

# Materials and Methods

### XLRP pedigree and pathogenic variant screening

The current study was designed according to the protocol of the Declaration of Helsinki and was approved by the local ethics

committee (Medical Ethical Committee of the Carl von Ossietzky University of Oldenburg, #2018-097; MHH ethical committee, #2576-2015). Written informed consent was obtained from all participants that were included in the study after the aims and consequences of the study had been explained to them in detail. The three-generation pedigree was prepared in line with the information provided by the family members. The pedigree included a male index patient, two affected females, one asymptomatic female, and two unaffected males. Peripheral blood samples and two independent fibroblast cell lines (A and B) were obtained from all study participants that are related to the family studied herein. In addition, two fibroblast cell lines from unrelated healthy females were included in this study. DNA was extracted from fibroblast cell lines and blood samples according to the manufacturer's recommendations using a Gentra Puregene kit (QIAGEN). DNA samples were quantified by Biospectrometer (Eppendorf) and Qubit Fluorometer (Thermo Fisher Scientific). The pathogenic variant in the gene *RPGR* (NM_000328.3, *RPGR*: c.1154T>A, p.Leu385*) previously was identified in the index patient (Vossing et al, 2020). The mutation was also identified in a heterozygous state in female carriers of the family using Sanger sequencing. The pathogenic *RPGR* variant was verified by Sanger sequencing in genomic DNA derived from blood and two independently generated biopsies of all the participants of the study. PCR amplification (primers and conditions are listed in Table S3) was applied with HotFire Tag polymerase (Solis Biodyne) according to the manufactures recommendations. Exo-SAP (New England Biolabs) treatments were used to purify the amplicons before Sanger sequencing. The BigDye Terminator v3.1 Cycle Sequencing Kit was applied on an ABI Prism 3130xl Genetic Analyzer (Applied Biosystem). The sequencing profile of each participant was compared with the *RPGR* reference sequence NM_000328.3 using Snap Gene software (GSL Biotech LLC).

## Clinical characterization of family members

All family members underwent a full ophthalmological examination including BCVA, Goldmann visual field (isopters I.1, I.2, I.3, III.4, and, if necessary, V.4), anterior segment slit lamp examination, and fundoscopy. Moreover, patients underwent multimodal imaging including ultra-widefield imaging (Optos California; Optos PLC), 55° FAF (Heidelberg Retina Angiography 2; Heidelberg Engineering), and spectral domain optical coherence tomography (Spectralis optical coherence tomography; Heidelberg Engineering) of the central retina. Affected family members (I:2, II:2, and III:1) as well as the asymptomatic female carrier (II:4) also underwent ffERG and mfERG examinations according to ISCEV standards (Hoffmann et al, 2021; Robson et al, 2022).

## Cell culture

Two fibroblast cell lines (A and B) were generated from two independently obtained skin biopsies from each of the following individuals: The index male patient, the affected female carriers, the asymptomatic female carrier, and the healthy male relatives. The skin biopsies were cut into small pieces and transferred to a sterile culture flask following the guidelines from previously published protocols (Villegas & McPhaul, 2005; Glaus et al, 2011). Once the explants were attached to the flask by air drying, MEM

(Biowest) supplemented with 1.3% L-glutamine (Biowest), 0.8% antibiotic (Biowest), and 20% FBS (Biowest) was added and incubated at 37°C and 5% $CO_2$. Upon near confluency of the fibroblasts, they were transferred to a 75 $cm^2$ culture flask. Cells were passaged upon 80–90% confluency and maintained in cell culture for propagation and analysis.

## *RPGR* transcript analyses

Before RNA extraction, $0.5 \times 10^6$ fibroblasts were seeded in a 25 $cm^2$ flask and maintained in a MEM growth medium (Biowest) for 24 h. Afterward, cells were starved for 48 h in a starvation medium (MEM medium [Biowest] supplemented with 1.75% L-glutamine [Biowest] and 1.25% antibiotic [Biowest]). RNA isolation was performed by the NucleoSpin RNA kit (Macherey-Nagel) according to the manufacturer's instructions. For cDNA synthesis, 500 ng RNA was transcribed using random primers and SuperScript III Reverse Transcriptase (Invitrogen). RT–PCR was performed to analyze the *RPGR* transcript (primers and conditions are listed in Table S3). *Vinculin* transcripts were used as an internal control.

### Immunocytochemistry and microscopy

Immunocytochemistry was applied to $0.15 \times 10^6$ fibroblasts on a 12 mm coverslip following a published protocol (Vossing et al, 2020). The cells were cultured in a 12-well plate and maintained in a MEM growth medium (Biowest) for 24 h after seeding. To induce cilia, cells were then cultured for 48 h in a starvation medium (MEM medium [Biowest] supplemented with 1.75% L-glutamine [Biowest] and 1.25% antibiotic [Biowest]). Cells were fixated with 4% PFA (Carl Roth) for 20 min followed by washing with PBS-T (1% PBS [Chemsolute] containing 0.05% Tween20 [AppliChem]) and incubation with 0.1 M Tris–HCL (pH 9.0) at 80°C for 30 min. Washing was performed again with PBS-T, followed by blocking unspecific antibody binding in PBS-T containing 2% BSA (fraction V; Carl Roth) for 30 min at room temperature. We used a sequential protocol, starting with a pair of primary and secondary antibodies, followed by a second pair of primary and secondary antibodies. Primary antibodies either were anti-RPGR antibody (1:200, HPA001593; Sigma-Aldrich), anti-gamma tubulin (1:500, ab11316; Abcam), anti-detyrosinated tubulin (1:500, ab3201; Merck, Millipore), or anti-polyglutamylation tubulin (GT335, 1:500; Bimol, AdipoGen Life Sciences). Secondary antibodies were Alexa-flour 488 (Rabbit, 1:2,000; Life Technologies), Alexa-flour 488 (Mouse, 1:2,000; Life Technologies), or Alexa-flour 568 (Rabbit, 1:2,000; Life Technologies). Cells were mounted with Fluoromount containing 4′,6-diamidino-2-phenylindole (DAPI) (Southern Biotech) for nuclei staining. A fluorescence microscope (Axiophot; Carl Zeiss) equipped with an AxioCam camera (Carl Zeiss) and Axiovision software (Carl Zeiss) was used to image the immunocytochemistry results. Ciliary axoneme length was measured from 250 cilia from five independent experiments using Fiji-ImageJ software (ImageJ).

### XCI measurements

XCI ratios were obtained from the genomic DNA extracted from either fibroblasts or blood samples. The samples were derived from

two affected carrier females (I:2 & II:2) and one asymptomatic carrier female (II:4). The methylation status of polymorphic repeats at four different X-chromosomal loci (androgen receptor [*AR*], *RP2*, SLIT and NTRK-like family member 4 [*SLITRK4*], proprotein convertase subtilisin/kexin type 1 inhibitor [*PCSK1N*]) was used to analyze the XCI ratio (active versus inactive allele). 300 ng genomic DNA was digested with the methylation-sensitive enzyme *HpaII* (50,000 U/ml, Cat number: R0171M; New England Biolabs) at 37°C for 16 h. *HpaII* will digest the unmethylated allele (active allele) at polymorphic regions of *AR*, *RP2*, *SLITRK4*, and *PCSK1N*. Polymorphic regions of *AR*, *RP2*, *SLITRK4*, and *PCSK1N* were amplified from *HpaII* digested and *HpaII* undigested DNA using FAM-labeled primers under similar PCR conditions (Table S3). Amplified PCR products were mixed with ROX100 size standard (Rox Ladder 50–1,000 bp, Cat number: 10-80-40 I; Biostep) and Hi-Di Formamide (Cat number: 4311320; Applied Biosystems) and were run on an ABI Prism 3130xl Genetic Analyzer (Applied Biosystem). The results were analyzed using the GeneMapper Software (GeneMapper v4.0; Applied Biosystems). The area under the peak of the digested samples was normalized with the area under the peak of the undigested samples to quantify the allele ratios. We used genomic DNA from the mother (I:2) as a reference to determine those XCI marker alleles (in the genes *AR*, *RP2*, *SLITRK4*, and *PCSK1N*) that were in phase with the reference or mutated-*RPGR* alleles in the daughters (II:2 and II:4). The analysis of the mother and their two daughters enabled us to distinguish the mutated allele from the reference allele. We used a threshold of ≥80:20 to define XCI skewing and ratios between 80:20 and 70:30 to define moderately skewed allele ratios. A ratio <70:30 was considered to represent a random XCI between both alleles.

## Statistics

Data are shown as means ± SD, if not stated otherwise. Data were analyzed by one-way ANOVA followed by Tukey's multiple comparison test using GraphPad Prism (GraphPad Prism 9.4.1, www.graphpad.com). Differences between the groups were considered non-significant if $P \geq 0.05$ and considered significantly different if $P < 0.05$.

## Data Availability

The data that support the findings of this study are available from the corresponding author upon reasonable request.

## Supplementary Information

## Acknowledgements

We are grateful to the family members for their donation of skin biopsies and blood samples. We further thank Dr. Jennifer A Lee from Greenwood Genetic Center, Greenwood, USA, for help in establishing the XCI assay. This work was funded by the Deutsche Forschungsgemeinschaft Research Training Group 1885 to J Neidhardt.

## Author Contributions

M Usman: data curation, software, formal analysis, validation, investigation, visualization, methodology, and writing—original draft.
C Jüschke: formal analysis.
F Song: methodology.
D Kastrati: methodology.
M Owczarek-Lipska: investigation and methodology.
J Eilers: methodology.
L Pauleikhoff: writing—original draft and writing—review and editing.
C Lange: funding acquisition, and writing—original draft, review, and editing.
J Neidhardt: conceptualization, resources, supervision, funding acquisition, project administration, and writing—review and editing.

## Conflict of Interest Statement

The authors declare that they have no conflict of interest.

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
