## [Reviewer comments · Life Science Alliance]

Life Science Alliance

Skewed X-inactivation is associated with retinal dystrophy in female carriers of RPGR mutations.

Muhammad Usman, Christoph Jüschke, Fei Song, Dennis Kastrati, Marta Owczarek-Lipska, Jannis Eilers, Laurenz Pauleikhoff, Clemens Lange, and John Neidhardt

DOI: <https://doi.org/10.26508/lsa.202201814>

Corresponding author(s): John Neidhardt, Medical Faculty, School of Medicine and Health Sciences

Review Timeline:

Submission Date:	2022-11-09
Editorial Decision:	2023-01-09
Revision Received:	2023-06-06
Editorial Decision:	2023-06-23
Revision Received:	2023-07-18
Accepted:	2023-07-24

Scientific Editor: Novella Guidi

Transaction Report:

January 9, 2023

Re: Life Science Alliance manuscript #LSA-2022-01814-T

John Neidhardt

Dear Dr. Neidhardt,

Thank you for submitting your manuscript entitled "Skewed X-inactivation is associated with RPGR ciliary mislocalization and retinal dystrophy in female mutation carriers" to Life Science Alliance. The manuscript was assessed by expert reviewers, whose comments are appended to this letter. We invite you to submit a revised manuscript addressing the Reviewer comments.

Thank you for this interesting contribution to Life Science Alliance. We are looking forward to receiving your revised manuscript.

Sincerely,

B. MANUSCRIPT ORGANIZATION AND FORMATTING:

Reviewer #1 (Comments to the Authors (Required)):

The work "Skewed X-inactivation is associated with RPGR ciliary mislocalization and retinal dystrophy in female mutation carriers" presented by Usman et al. aims to explain why heterozygous carrier females of a pathogenic null mutation in RPGR (the major gene causative of X-linked Retinitis Pigmentosa XL-RP) can show severe retinal alterations.

In this work, the authors have characterized several primary fibroblast cell lines obtained from a three generation family with one male and several affected females. They have strived to demonstrate that skewed X-chromosome inactivation is the cause of the RP phenotype in females, and to this end, they have characterized RPGR transcript levels in these lines, studied RPGR localization in the primary cilium, and determined ciliary length. Besides, they have performed analysis of X-chromosome inactivation and associated the observed phenotype to the detected X-chromosome inactivation pattern.

The conclusions are relevant and interesting, particularly for clinical genetic diagnosis of IRD patients, in which potential new mutations and VUS identified in RPGR have to be assigned, but also, to provide a rationale for the phenotypic alterations observed in many heterozygous carrier females of X-linked genes. Other conclusions that are worth for other scientists working in X-linked genes concern the increase in the number of samples used to study X-chromosome inactivation. Since neurological tissues such as the retina cannot be sampled from patients, other proxy tissues must be analysed to check out gene inactivation or gene expression. The authors conclude that the blood does not provide a good model to study X-chromosome inactivation, whereas several fibroblast samples must be analysed because there is variability in which X-chromosome is finally inactivated. Also, they propose that several X-chromosome genetic markers should be analysed to determine which chromosome is inactivated in the analysed samples.

All in all, the conclusions are valid, the work is well performed, the images are curated and the statistics are correct. N

MAIN POINT

I have a relevant concern regarding the controls used in the work presented. The authors decide to use internal controls within the family, in males they have affected and non-affected males. In this case, unaffected males do not carry the mutation they are studying. However, in females, they do not use any healthy female, but an asymptomatic carrier within the family. Indeed, this was already necessary to make the correlations between expression levels of RPGR and the observed phenotype, but I still consider that the work would have benefitted much from adding also a completely healthy homozygous wild-type female to compare, as an external reference, particularly for:

- Figures 2A and 2C, where the wild-type RPGR levels would have allowed to compare the results of the apparently healthy (although with some slight symptoms) female and her mother and sibling to a control female.
- Figure 5, in which the expected ciliary length in a wild-type female would have been a counterpart to the ciliary length in the several samples from heterozygous females. Samples A from the three analysed females have a similar result than the healthy wild-type males, yet two of them are affected and samples B show a distinct phenotype. It would have been wise to see whether a normal female showed a similar or a longer ciliary length.

Finally, the addition of a completely healthy female control would have provided a reference phenotypic information, very useful for the assignment of pathogenicity to VUS identified in RPGR.

MINOR POINTS

The text has some typos here and there that should be revised. Although this is not an exhaustive list, please check:

- line 163, it should read "analyzed"
- line 167, it should read "extent"
- line 171, it should read "detectable"
- throughout the text, please check out the greek letters as symbols, as delta, for instance the ciliary protein is delta-tubulin, not d-tubulin.
- line 269, it should read "tendency"
- line 297, it should read "X-chromosome"
- line 330, it should read "X-chromosome"
- line 369, it should read "functionally null mutation"
- lines 379 and 389. Please rephrase, a female does not present a male phenotype. A female may present a male-like phenotype, but I still think this is a dubious comparison. Probably it should read: a full RP phenotype, or a severe RP phenotype comparable to that presented by males.
- line 390, it should read "Despite the fact"

Reviewer #2 (Comments to the Authors (Required)):

In this manuscript, Usman and colleagues describe two female carriers affected by Retinitis pigmentosa and one asymptomatic female from a three-generation family segregating a nonsense pathogenic variant in RPGR gene. The proband had been previously reported (Vossing et al. 2020). To understand the contribution of X chromosome inactivation (XCI) skewing in the phenotype variability and differences in RPGR ciliary localization among the female carriers, they got two independent control and patient-derived fibroblast cell lines (A and B) from female carriers and from the index male patient, besides blood. As RPGR is subject to XCI, they analysed four X-linked loci on the X chromosome that have previously been described as markers for skewed XCI. The subject is interesting, mainly concerning that RPGR pathogenic variants affect males in particular, but some points should be clarified.

Major points:

- Lines 38-40: "Taken together, our data strongly suggests that skewed XCI causes ciliary dysfunction and clinical phenotypes in female carriers of RPGR mutations".

Please rewrite this sentence and other with the same meaning along the text (ex. Lines 289-290). Skewed XCI should not be pointed as a cause of a clinical phenotype. Secondary skewing occurs as a post-inactivation cell selection, acting for or against cells carrying one particular X chromosome. In fact, it is the likely pathogenic variant in RPGR that caused the clinical phenotype and XCI could just modulate the phenotype variability. The affirmative should be more nuanced in the whole text.

- Lines 56/57: "More than 60 genes have been reported to cause RP (Stone 57 et al. 2017)". Please, review this information. Human Phenotype Ontology lists much more genes associated to RP.

- Lines 104-106: "In this study, we investigated females with RPGR-associated XLRP and found that skewed XCI is associated with ciliary mislocalization of RPGR, shortening of primary cilia and RP".

Authors should clarify that this is not an investigation study of females carrying RPGR pathogenic variants, but a family study, in which female carriers harbouring the same variant with similar genetic background were investigated.

- Lines 292-304: "To verify whether a non-random XCI may explain the differences in RPGR protein localization along cilia, we analyzed the methylation status between the two X-chromosomal alleles in females family members. We compared cell lines A and B of female RPGR mutation carriers and analyzed four loci on the X chromosome that have previously been described as markers for skewed XCI. We found that cell line A of the affected carrier female I:2 showed a skewed methylation of X-chromosomal alleles, which is causing the expression of the reference-RPGR allele (Table 1)...."

The 5mCpG-sensitive restriction endonuclease-based PCR assay targeting the polymorphic nucleotide tandem repeat in the X chromosome is a standard readout method for determining the methylation statuses of alleles on X_a and X_i and is widely used as a marker of X-chromosome activity. However, the method used for exploring four loci previously described as markers for skewed XCI is not sufficient to explore which RPGR allele (reference or alternative) is expressed. Indeed, even that the authors used the phase according to the grandmother genotypes, this information could only be accessed if cDNA samples from the patients were sequenced in a semi quantitative manner. It could be very informative if the authors could sequence the cDNA of the two symptomatic females and the asymptomatic one for comparative purposes.

- Line 327, table 1:

The authors stated that they used the mean of triplicates, but how do the authors explain the differences in XCI ratios between cell lines in the same individual? As an example, please see RP2 and SLITRK4 markers for individual II.2 cell lines A and B. Other question is: how old are the female carriers studied? It is vastly known that XCI skewing naturally increases after 35 years old.

Which was the criteria used for the interpretation of the results as "ref-RPGR expressed", "mut-RPGR expressed", "mut-RPGR expressed" or "unclear"?

The number in red is not the only one that do not correlate with other markers (see PCSK1N for individual II.2 A).

- Lines 330-331: "Footnote: Methylation of X-chromosomal alleles (encoding RPGR) was considered extremely skewed towards either allele if ratios of XCI were {greater than or equal to}80:20". Generally, extreme skewing is represented by a ratio of >90:10.

- Which are the authors hypotheses for an unfavourable XCI skewing in the symptomatic female carriers? Existence of a second X-linked pathogenic variant that provoked XCI skewing, forcing the expression of the alternative allele in an RPGR gene on the active X chromosome? Please see DOI: 10.1007/s12035-020-01981-8 for improving discussion.

- Authors should consider that cell cultures could have also an impact in XCI patterns.

- Lines 525-527: authors should state the reference they used to calculate XCI ratios and which intervals were used to define XCI skewing in the methodology.

Minor points:

- Replace the word "mutation" by "pathogenic variant", according to current nomenclature.

- Line 31/32: "To understand the molecular mechanism underlying RP in female RPGR mutation" - replace "in female RPGR mutation" by "in female carriers of RPGR pathogenic variants".

- Line 32: replace "non-sense" by "nonsense" in the whole text.

- Line 118: describe BCVA, FAF, FfERG and mfERG

-Line 163: analyzed

- Please, review some type errors along the text.

Authors response to reviewer #1**Reviewer #1**

The work "Skewed X-inactivation is associated with RPGR ciliary mislocalization and retinal dystrophy in female mutation carriers" presented by Usman et al. aims to explain why heterozygous carrier females of a pathogenic null mutation in RPGR (the major gene causative of X-linked Retinitis Pigmentosa XL-RP) can show severe retinal alterations.

In this work, the authors have characterized several primary fibroblast cell lines obtained from a three generation family with one male and several affected females. They have strived to demonstrate that skewed X-chromosome inactivation is the cause of the RP phenotype in females, and to this end, they have characterized RPGR transcript levels in these lines, studied RPGR localization in the primary cilium, and determined ciliary length. Besides, they have performed analysis of X-chromosome inactivation and associated the observed phenotype to the detected X-chromosome inactivation pattern. The conclusions are relevant and interesting, particularly for clinical genetic diagnosis of IRD patients, in which potential new mutations and VUS identified in RPGR have to be assigned, but also, to provide a rationale for the phenotypic alterations observed in many heterozygous carrier females of X-linked genes. Other conclusions that are worth for other scientists working in X-linked genes concern the increase in the number of samples used to study X-chromosome inactivation. Since neurological tissues such as the retina cannot be sampled from patients, other proxy tissues must be analysed to check out gene inactivation or gene expression. The authors conclude that the blood does not provide a good model to study X-chromosome inactivation, whereas several fibroblast samples must be analysed because there is variability in which X-chromosome is finally inactivated. Also, they propose that several X-chromosome genetic markers should be analysed to determine which chromosome is inactivated in the analysed samples.

All in all, the conclusions are valid, the work is well performed, the images are curated and the statistics are correct. N

MAIN POINT

I have a relevant concern regarding the controls used in the work presented. The authors decide to use internal controls within the family, in males they have affected and non-affected males. In this case, unaffected males do not carry the mutation they are studying. However, in females, they do not use any healthy female, but an asymptomatic carrier within the family. Indeed, this was already necessary to make the correlations between expression levels of RPGR and the observed phenotype,

but I still consider that the work would have benefitted much from adding also a completely healthy homozygous wild-type female to compare, as an external reference, particularly for:

- Figures 2A and 2C, where the wild-type RPGR levels would have allowed to compare the results of the apparently healthy (although with some slight symptoms) female and her mother and sibling to a control female.

- Figure 5, in which the expected ciliary length in a wild-type female would have been a counterpart to the ciliary length in the several samples from heterozygous females. Samples A from the three analysed females have a similar result than the healthy wild-type males, yet two of them are affected and samples B show a distinct phenotype. It would have been wise to see whether a normal female showed a similar or a longer ciliary length.

Finally, the addition of a completely healthy female control would have provided a reference phenotypic information, very useful for the assignment of pathogenicity to VUS identified in RPGR.

We very much appreciate your time and effort in reviewing our manuscript and providing us with your helpful feedback. Your concern regarding the controls used in our study is well noted, and we agree that including a completely healthy homozygous wild-type female control would provide additional useful data. We now have included two fibroblast cell lines derived from healthy females, individuals that are unrelated to the family presented in our manuscript. The new family-independent/unrelated controls did not agree to be evaluated by fundus and/or ERG measurements, as they did not report any signs of visual impairments, nor did show sequence alterations in RD-associated genes. Instead, we cite publications where unaffected individuals are phenotypically described in detail (line 403,-404).

In accordance with your suggestion, we now did compare the new healthy controls to an affected family member. We performed three independent replicates for the following datasets:

- (i) RPGR transcript expression (supplementary Fig S1 A-B),
- (ii) RPGR localization along the cilium (supplementary Fig S2 A-B)
- (iii) cilium length measurements (supplementary Fig S3 A-B).

In both of these family-independent controls, we found reference levels of RPGR transcripts, RPGR localization, and cilium length. These data were similar to cell line A of 1:2, but significantly different from cell line B. In summary, the new data provides additional support to our findings as described in the first version of the manuscript. We have revised our manuscript accordingly in line 142-147, 181-188, 237-238, 245-246, 278-284, 321-324)

MINOR POINTS

The text has some typos here and there that should be revised. Although this is not an exhaustive list, please check:

Thank you for your suggestions. We corrected the following typos, unless otherwise noticed.

-line 163, it should read "analyzed"

-line 167, it should read "extent"

-line 171, it should read "detectable"

-throughout the text, please check out the greek letters as symbols, as delta, for instance the ciliary protein is delta-tubulin, not d-tubulin.

Thank you for your comment. However, the ciliary protein referred to in our study is detyrosinated alpha tubulin, commonly written as d-tubulin, rather than delta-tubulin. We have verified the symbol usage throughout the text and have made sure that it is consistent with the current convention.

-line 269, it should read "tendency"

-line 297, it should read "X-chromosome"

-line 330, it should read "X-chromosome"

-line 369, it should read "functionally null mutation"

- lines 379 and 389. Please rephrase, a female does not present a male phenotype. A female may present a male-like phenotype, but I still think this is a dubious comparison. Probably it should read: a full RP phenotype, or a severe RP phenotype comparable to that presented by males.

-line 390, it should read "Despite the fact"

Authors response to reviewer #2

Reviewer #2

In this manuscript, Usman and colleagues describe two female carriers affected by Retinitis pigmentosa and one asymptomatic female from a three-generation family segregating a nonsense pathogenic variant in RPGR gene. The proband had been previously reported (Vossing et al. 2020). To understand the contribution of X chromosome inactivation (XCI) skewing in the phenotype variability and differences in RPGR ciliary localization among the female carriers, they got two independent control and patient-derived fibroblast cell lines (A and B) from female carriers and from the index male patient, besides blood. As RPGR is subject to XCI, they analysed four X-linked loci on the X chromosome that have previously been described as markers for skewed XCI. The subject is interesting, mainly concerning that RPGR pathogenic variants affect males in particular, but some points should be clarified.

We would like to thank the reviewer for the helpful suggestions to our manuscript! We very much appreciate the efforts and time invested.

In the following, we provide point by point answers.

Major points:

- Lines 38-40: "Taken together, our data strongly suggests that skewed XCI causes ciliary dysfunction and clinical phenotypes in female carriers of RPGR mutations".

Please rewrite this sentence and other with the same meaning along the text (ex. Lines 289-290). Skewed XCI should not be pointed as a cause of a clinical phenotype. Secondary skewing occurs as a post-inactivation cell selection, acting for or against cells carrying one particular X chromosome. In fact, it is the likely pathogenic variant in RPGR that caused the clinical phenotype and XCI could just modulate the phenotype variability. The affirmative should be more nuanced in the whole text.

Thank you for your comment. We agree and have revised our manuscript accordingly (Line 43-45, 305, 370-371, 453)

- Lines 56/57: "More than 60 genes have been reported to cause RP (Stone 57 et al. 2017)". Please, review this information. Human Phenotype Ontology lists much more genes associated to RP.

Thank you, we have updated our manuscript. We chose to cite the number of genes listed for non-syndromic RP from the RetNet database. We believe that the category of non-syndromic RP matches best with the phenotype of our family. Indeed, those over 220 genes listed in the HPO database also include syndromes and RP-resembling phenotypes. To make our manuscript more precise/understandable, we added 'non-syndromic' and corrected the sentences and citations (Line 59-60).

- Lines 104-106: "In this study, we investigated females with RPGR-associated XLRP and found that skewed XCI is associated with ciliary mislocalization of RPGR, shortening of primary cilia and RP". Authors should clarify that this is not an investigation study of females carrying RPGR pathogenic variants, but a family study, in which female carriers harbouring the same variant with similar genetic background were investigated.

Thank you for your valuable feedback. We appreciate your suggestion and have revised the sentence accordingly. Our study is a family study, in which we investigated female carriers of the same RPGR pathogenic variant with similar genetic backgrounds. We have edited the sentence accordingly (Line 108-109).

- Lines 292-304: "To verify whether a non-random XCI may explain the differences in RPGR protein localization along cilia, we analyzed the methylation status between the two X-chromosomal alleles in females family members. We compared cell lines A and B of female RPGR mutation carriers and analyzed four loci on the X chromosome that have previously been described as markers for skewed XCI. We found that cell line A of the affected carrier female I:2 showed a skewed methylation of X-chromosomal alleles, which is causing the expression of the reference-RPGR allele (Table 1)...." The 5mCpG-sensitive restriction endonuclease-based PCR assay targeting the polymorphic nucleotide tandem repeat in the X chromosome is a standard readout method for determining the methylation statuses of alleles on X_a and X_i and is widely used as a marker of X-chromosome activity. However, the method used for exploring four loci previously described as markers for skewed XCI is not sufficient to explore which RPGR allele (reference or alternative) is expressed. Indeed, even that the authors used the phase according to the grandmother genotypes, this information could only be accessed if cDNA samples from the patients were sequenced in a semi quantitative manner. It could be very informative if the authors could sequence the cDNA of the two symptomatic females and the asymptomatic one for comparative purposes.

Thank you for your insightful comments. In accordance with your suggestion and to exclude a mis-interpretation from the XCI table in our manuscript (Table1), we now generated a new summary table (Table S3).

Furthermore, we performed Sanger sequencing of a cDNA fragment that included the RPGR pathogenic variant (Fig S1 C). In line with our interpretation, we found that cell line A of affected female carriers predominantly expressed Ref-RPGR, while cell line B of the affected female carriers expressed Mut-RPGR. And both cell lines A and B of asymptomatic carrier female predominantly express Ref-RPGR. In addition, we also included two unrelated female controls and confirm that they also express Ref-RPGR. We have revised our manuscript accordingly (Line 145-147, 181-188, 321-324).

- Line 327, table 1:

The authors stated that they used the mean of triplicates, but how do the authors explain the differences in XCI ratios between cell lines in the same individual? As an example, please see RP2 and SLITRK4 markers for individual II.2 cell lines A and B.

Thank you for your feedback on our XCI analysis. We would like to clarify that we used the mean values of XCI ratios from three independent repeats. The mean \pm SD is now given in Table 1. We found these ratios to be consistent and reproducible across the replicates (please also compare to supplementary table S2).

Regarding the differences in XCI ratios between cell lines A and B of the same individual, we would like to explain that cell line A and B are derived from two different areas of the skin from the same individual, and, as females are mosaic for XCI, it rather is expected that different parts of a tissues show different XCI patterns (as observed in our dataset).

Other question is: how old are the female carriers studied? It is vastly known that XCI skewing naturally increases after 35 years old.

The ages of the female participants in our study are as follows: I:2=60 years, II:2=42 years, and II:4=40 years. Although it is known that XCI skewing increases with age in blood tissues (Busque et al. 2009), we did not find clear indications for this in our study. Since our study did not longitudinally follow the females to fully assess age-related effects. To mitigate this limitation, we utilized independently skin-derived fibroblast cell lines A and B and observed consistent XCI ratios across multiple passages. While we cannot rule out the potential influence of age, our findings rather suggest stability in XCI skewing within the studied cell lines in the cell culture (Supplementary table S2). The only hint in our dataset that might point towards a higher skewedness with higher age: The grandmother showed the highest level of skewedness compare to the other family members; nevertheless, this observation might also be coincidence.

We now added a paragraph in the discussion to mention the possibility of increasing skewedness with age also to the readers (line428-436).

Which was the criteria used for the interpretation of the results as "ref-RPGR expressed", "mut-RPGR expressed", "mut-RPGR expressed" or "unclear"?

Based on our dataset from cell lines A and B, we were able to interpret the expressed allele for each cell line by correlating with the XCI ratios. However, upon careful consideration, we recognized that the interpretation provided in Table 1 may be unclear to readers. Consequently, we have made the decision to remove the interpretation from table 1. And in order to enhance the clarity we made another table to summarize our findings for the ease of the reader (supplementary table S3).

The number in red is not the only one that do not correlate with other markers (see PCSK1N for individual II.2 A).

We agree and highlighted the PCSK1N for individual II.2 A as well. The highlight is described in the foot note of the table 1 (line 360-362).

- Lines 330-331: "Footnote: Methylation of X-chromosomal alleles (encoding RPGR) was considered extremely skewed towards either allele if ratios of XCI were {greater than or equal to}80:20". Generally, extreme skewing is represented by a ratio of >90:10.

Thank you for your comments. We appreciate your input regarding the threshold for extreme skewing of XCI ratios. We changed our wording from "extremely skewed" to "skewed" to avoid misunderstandings. While we acknowledge that some studies have used a threshold of

>90:10 to define extreme XCI skewing, thresholds of greater than or equal 80:20 regularly are also defined as extremely skewed XCI (Fahim et al. 2020; González-Ramos et al. 2020). Further, we believe that the criteria we described for moderately skewed and skewed XCI ratios are justified by our observations and thus should be considered valid. We hope that this clarifies our rationale for using this threshold in our study. We have revised our manuscript accordingly and added a paragraph in the discussion (line 444-450, 574-577).

- Which are the authors hypotheses for an unfavorable XCI skewing in the symptomatic female carriers? Existence of a second X-linked pathogenic variant that provoked XCI skewing, forcing the expression of the alternative allele in an RPGR gene on the active X chromosome? Please see DOI: 10.1007/s12035-020-01981-8 for improving discussion.

Thank you for your valuable comments on our study. We now refer to DOI: 10.1007/s12035-020-01981-8 and improved the discussion in line 437-443.

- Authors should consider that cell cultures could have also an impact in XCI patterns.

Thank you for your comment. To minimize this potential influence, we made sure to maintain consistent cell culture conditions such as passaging, handling, and freezing for all the cell lines used in our study. We also repeated our experiments independently, and our results were reproducible (Supplementary table 2). Nevertheless, we cannot completely exclude this possibility and added a sentence to the discussion to express this concern (Line 432-436).

- Lines 525-527: authors should state the reference they used to calculate XCI ratios and which intervals were used to define XCI skewing in the methodology.

Thank you for your helpful comment. In our study, we utilized the mother (I:2) as a reference to determine which allele of the XCI markers (AR, RP2, SLITRK4, and PCSK1N) was in phase with the reference versus mutated RPGR allele in the daughters (II:2 and II:4). By analyzing the heterozygous mother and daughters, we were able to distinguish the mutated allele from the reference allele (Line 570-574).

Minor points:

- Replace the word "mutation" by "pathogenic variant", according to current nomenclature.

Acknowledged and corrected.

- Line 31/32: "To understand the molecular mechanism underlying RP in female RPGR mutation" - replace "in female RPGR mutation" by "in female carriers of RPGR pathogenic variants".

Acknowledged and corrected.

- Line 32: replace "non-sense" by "nonsense" in the whole text.

Acknowledged and corrected.

- Line 118: describe BCVA, FAF, FfERG and mfERG

Acknowledged and corrected.

analysed-Line 163: anlyzed

Acknowledged and corrected.

- Please, review some type errors along the text

Acknowledged and corrected.

References

- Busque L, Paquette Y, Provost S, Roy DC, Levine RL, Mollica L, Gilliland DG. 2009. Skewing of X-inactivation ratios in blood cells of aging women is confirmed by independent methodologies. *Blood* **113**: 3472-3474.
- Fahim AT, Sullivan LS, Bowne SJ, Jones KD, Wheaton DKH, Khan NW, Heckenlively JR, Jayasundera KT, Branham KH, Andrews CA et al. 2020. X-Chromosome Inactivation Is a Biomarker of Clinical Severity in Female Carriers of RPGR-Associated X-Linked Retinitis Pigmentosa. *Ophthalmol Retina* **4**: 510-520.
- González-Ramos IA, Mantilla-Capacho JM, Luna-Záizar H, Mundo-Ayala JN, Lara-Navarro IJ, Ornelas-Ricardo D, González Alcázar J, Evangelista-Castro N, Jaloma-Cruz AR. 2020. Genetic analysis for carrier diagnosis in hemophilia A and B in the Mexican population: 25 years of experience. *Am J Med Genet C Semin Med Genet* **184**: 939-954.

June 23, 2023

RE: Life Science Alliance Manuscript #LSA-2022-01814-TR

Prof. John Neidhardt
University Oldenburg
Germany

Dear Dr. Neidhardt,

Thank you for submitting your revised manuscript entitled "Skewed X-inactivation is associated with retinal dystrophy in female carriers of RPGR mutations.". We would be happy to publish your paper in Life Science Alliance pending final revisions necessary to meet our formatting guidelines.

- please upload all figure files as individual ones, including the supplementary figure files; all figure legends should only appear in the main manuscript file after the references section.
- please make sure the manuscript sections are aligned in accordance with LSA's formatting guidelines: please separate the Figure legends and Supplemental Figure legends into separate sections
- Graphical abstract please upload with file designation "Graphical Abstract."
- please add ORCID ID for the corresponding author--you should have received instructions on how to do so
- please make sure the author order in your manuscript and our system match
- in the manuscript text, there is a reference to Figure S4, but the figure itself is not included

A. FINAL FILES:

B. MANUSCRIPT ORGANIZATION AND FORMATTING:

Sincerely,

Reviewer #1 (Comments to the Authors (Required)):

I believe that this work is not only interesting for researchers involved in the genetics of inherited blindness, but also provides a framework for validating X-skewed inactivation in cells from carrier females who also show a phenotype associated to mutations in genes located in the X chromosome. This topic has been a difficult issue in X-linked diseases and the potential phenotypic impact in female carriers.

The authors have adequately addressed my previous concerns and have added data from control females as suggested, which further validates their results. This new version also addresses the comments by other reviewers. Therefore, I recommend it for publication.

Reviewer #2 (Comments to the Authors (Required)):

Upon receipt of the revised version of the manuscript entitled "Skewed X-inactivation is associated with retinal dystrophy in female carriers of RPGR mutations (LSA-2022-01814-TR)", I consider that overall, the authors have satisfactorily addressed the inquiries posed by myself as well as the additional reviewer. Consequently, I believe that now the manuscript has attained the requisite level of suitability for publication in the Life Science Alliance.

July 24, 2023

RE: Life Science Alliance Manuscript #LSA-2022-01814-TRR

John Neidhardt
Medical Faculty, School of Medicine and Health Sciences
Human Genetics, Medical Faculty, School of Medicine and Health Sciences
Carl von Ossietzky Universität Oldenburg
Oldenburg 26129
Germany

Dear Dr. Neidhardt,

Thank you for submitting your Research Article entitled "Skewed X-inactivation is associated with retinal dystrophy in female carriers of RPGR mutations.". It is a pleasure to let you know that your manuscript is now accepted for publication in Life Science Alliance. Congratulations on this interesting work.

DISTRIBUTION OF MATERIALS:

Again, congratulations on a very nice paper. I hope you found the review process to be constructive and are pleased with how the manuscript was handled editorially. We look forward to future exciting submissions from your lab.

Sincerely,
